# Analysis of Ocean Parameters as Sources of Coastal Storm Damage: Regional Empirical Thresholds in Northern Spain

**Victoria Rivas \*, Carolina Garmendia and Domingo Rasilla** 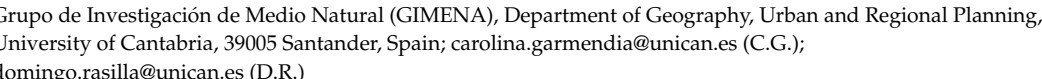

Grupo de Investigación de Medio Natural (GIMENA), Department of Geography, Urban and Regional Planning, University of Cantabria, 39005 Santander, Spain; carolina.garmendia@unican.es (C.G.); domingo.rasilla@unican.es (D.R.)

\* Correspondence: maria.rivas@unican.es

**Abstract:** This contribution aims to explore the role of oceanographic parameters on the damage caused by storms at the eastern Cantabrian coast (1996–2016). All wave storms affecting the study area were characterized in terms of several oceanographic parameters; among them, damaging storms (responsible for direct and tangible loss) were identified. Cross-referencing both databases makes it possible to find some thresholds that explain storm conditions associated with property damage. Particularly relevant are those responsible for significant and widespread damage: maximum significant offshore wave height >6.5 m, maximum total water level >6 m, SPI > 1700 m$^2$h, and a storm duration >48 h. These values are exceptionally high, mostly exceeding the 95th percentile. A comparison has been made with other thresholds described in the literature. The concurrence of high wave height and high tidal level is crucial as the greatest damage is caused by the combination of wave impact and over-wash, so a long duration of the storm is necessary to coincide with high tide. An empirical Intensity-Duration threshold has also been obtained with the following function $I = 248.7 \, D^{-0.45}$. Damage can occur with moderate storms, but with severe effects only with exceptional wave and sea-level values, during long-lasting storms.

**Keywords:** damaging storm; direct damage; marine storm thresholds; eastern Cantabrian coast





## 1. Introduction

Atmospherically induced extreme wave storms are one of the main hazards in coastal regions. Their impacts range from sudden and dramatic changes in coastal morphologies to destructive effects on human properties and lives. Such impacts result from the huge amount of energy released in short time periods, synergistically combining different dynamic agents: deep low pressures trigger high wind speeds pushing the ocean surface up in the form of storm surges, while high waves over-wash the hinterland when they encounter the coast. On the other hand, the worldwide trend toward the concentration of population and economic activities (residential, agricultural, fisheries, industries, tourism, etc.) on coastal areas is increasing the amount and value of properties exposed to such events [1].

While the general population perceives these episodes as exceptional and, in many cases, their consequences as inevitable, they are a recurrent phenomenon that offers the possibility to take palliative measures aimed to reduce their impacts. Consequently, there is a need to progress into the knowledge of the characteristics of the storm conditions, responsible for widespread impacts along the coast. In addition, the incorporation of socioeconomic information is essential in any risk assessment and constitutes the basis of the decision-making process on mitigation measures [2–7].

In Europe, wave storms are responsible for about half of the economic losses attributed to natural disasters, comparable in terms of potential damage to hurricanes in the U.S.A. or to earthquakes in Japan [8,9]. In this context, they have been an object of investigation

by researchers aiming to better understand their dynamics, effects, long-term trends, and future behavior at European coasts under climate change. The northern coast of Spain shares a similar exposure to the Atlantic storms since it has been hit by several exceptional and damaging storms (e.g., Lothar and Martin, December 1999; Klaus, January 2009; Xynthia, February 2010; Gong, January 2013, etc.) [10,11].

There are many references aimed at finding relationships between storm intensity and damage. Usually, the determination of storm intensity is carried out through the values of specific ocean-meteorological parameters (mainly significant wave height), or by indicators resulting from their combination. Other procedures widely used for defining storm strength are based on morphological and sedimentological coastal response (changes in dune or berm height and/or beach width) and over-wash occurrence, and much less on the effects of human settlement. Most of the methodologies are based on post-event field surveys and/or different documentary records from aerial photographs, technical reports of public agencies, regional newspapers, research papers, personal communication with neighbors, etc. Regarding the scale of analysis, some authors cover a very large study area and a very long time period [12], while others focus on a single storm event [13,14]. On the other hand, there are authors who combine time series at different scales for very specific locations [15,16]. There are even works that exclusively quantify the vulnerability of beaches to wave storms [17,18], or classify storms according to their intensity and their effects on beaches [19]. Finally, some focus on the determination of quantitative thresholds [20–22]. In northern Spain, they mainly focus on specific features or singular events [23–28] or on the long-term trends of storminess [29–39].

This paper presents an analysis of the causal relationships between hazard characteristics of the marine storms with the magnitude of the economic damages on the eastern edge of the Spanish Cantabrian coast (Cantabria and Basque Country), that is, at the regional scale, and for a long-time span, over the period 1996–2016. The main goal is to establish statistical links and solid correlations between the hydrodynamic forcing of wave storm climatology (set of values of different oceanographic variables or combinations of them) and magnitude and type of documented record of damage on infrastructures, real estate properties (dwellings, business offices, shops, etc.), and vehicles. In short, it is about determining the storm conditions that, due to their exceptional character, are likely to produce damages, especially, those capable of producing major and widespread damage at the coastal area.

We adopt a generic post-damage procedure through the indemnities paid by the reinsurance corporation; that is, storm damage is "measured" by quantifying its monetary effects. Although this quantification does not cover all effects produced by storms in a broad sense (it excludes environmental, health, and cultural impacts), it allows a robust assessment in terms of economic losses from post-event data [40].

On the other hand, the quantification of the damage caused by storms to infrastructure is very complex; although, it is always proportionally much more expensive than damage to other types of assets. The CCS points out that in this case, the ownership belongs to the public administrations (they are located in the Maritime–Terrestrial Public Domain—DPMT), and although the system allows them to be insured, the decision to do so varies according to the political criteria of each moment. In general, they tend to "self-insure"; such as, they repair it from their budgets (this is less widespread in local and autonomous administrations) [41,42], as happened in the Cantabrian in 2014, in 2015 in the Atlantic and Mediterranean slopes, and in 2018 in the Mediterranean [43]. In this type of damage, another aspect to consider is that these do not reflect the intensity of the storm due to what engineers refer to as "material fatigue". This is a progressive process that depends not only on the age of the infrastructure, but also on the "wear and tear" caused by changes in the thrusts to which it has been subjected over time (even when these may have been several times lower than the maximum for which it was designed). In short, these constructions have been able to withstand the impact of more intense storms and, due to their wear and tear, have been damaged by others of lower energy.

The reconstruction of the past occurrence entails the identification of the major episodes, as well as the assessment of the exceptional magnitude of them, through a set of indicators representative of the processes responsible for damage: wave impact and over-wash. The empirical regional thresholds obtained, defined as storm attributes (wave height, sea level, energy, etc.), which best explain the conditions that resulted in coastal economic damage, represent the minimum values beyond which monetary damage has occurred in the past, and therefore is likely to do in the future. It must be pointed out that the threshold values obtained in this work refer exclusively to extreme conditions that enable the simultaneous occurrence of a large amount of damage widespread over the study area and, therefore, should be associated with clearly extraordinary conditions; although, minor damage has obviously also occurred under less extreme conditions.

This kind of approach will allow the relative importance of each of the storm parameters on risk situations arising from these high-energy episodes to be discriminated. The determination of thresholds for storm-triggered coastal damage is one of the most effective methods for developing an early warning system, so the results obtained can be a good starting point to allow for reliable forecasts. More accurate predictions of storm impacts will be useful for the development of mitigation measures based on coastal planning and management aimed at minimizing damage.

## 2. Materials and Methods

### 2.1. Study Area

The studied area covers the northern coast of the Iberian Peninsula, with a total length of around 576 km [44], including 3 provinces (Cantabria, Vizcaya, and Guipúzcoa) (Figure 1).

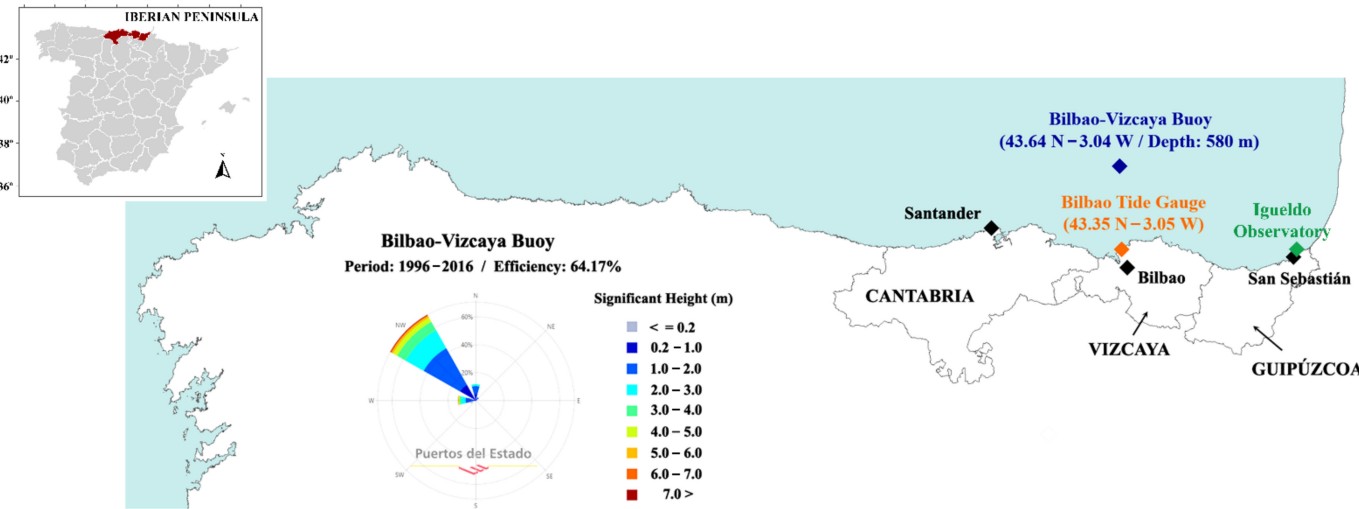

**Figure 1.** Study area and location of data sources (tide gauge, buoy, and meteorological observatory).

The coastline is straight, W–E oriented, and prolonged by a narrow continental shelf, which offers little protection to a coastline directly affected by wave energy. Shoreline is formed mainly of rocky cliffs (Figure 2); most of them composed of compact materials [45,46]. Beaches, mainly sandy, are short and narrow; some of them include dune fields. Low-lying coast around embayments, river mouths, and estuaries are the preferred places for the location of most of the urban areas, largely on reclaimed intertidal areas. Artificial elements are linked to port and defense structures (seawalls, jetties) and urban promenades.

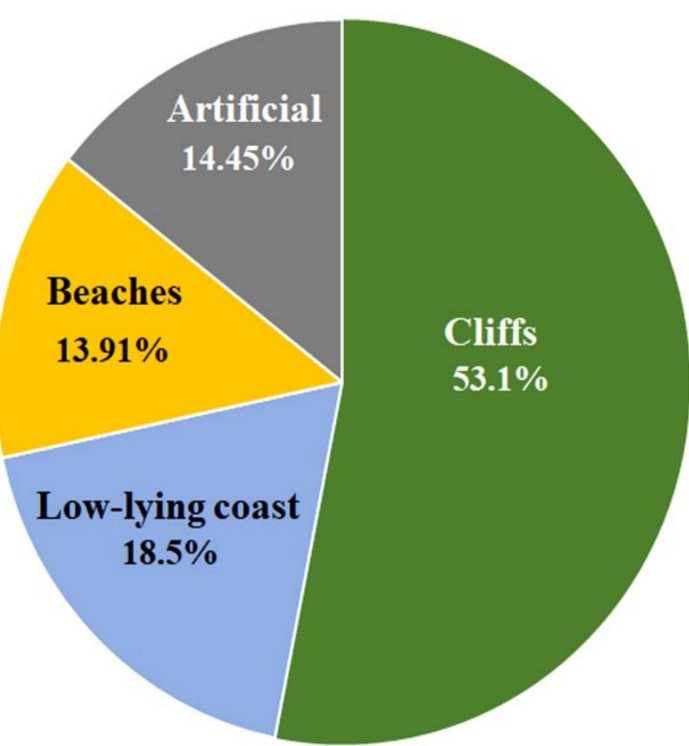

**Figure 2.** Coastal configuration of study area at regional scale. Date taken from [47].

The atmospheric and oceanographic dynamic in the north of the Iberian Peninsula is complex, and is very conditioned by the proximity to the coast (between 30 and 50 km) of a mountain alignment, which exceeds in many points 2000 m. This arrangement generates significant mesoscale disturbances in the direction and strength of winds during the crossing of extratropical cyclones. Several wave regimes have been found in our study area, closely related to the wind direction and strength and the length of the fetch [36,48,49]. Wave regimes in the Bay of Biscay show a seasonal modulation [32,33,50]. The winter mean sea state consists of a wave-swell reaching a significant wave height of about 2 to 3 m, and periods of 8–15 s, usually generated by storms tracking further north the 60 °N parallel [51].

Episodes of marine storms are linked to deep lows embedded into the dominant zonal circulation tracking through the British Isles (between 50 and 60 °N). Storm conditions produce significant wave heights of more than 7 m, reaching 11 m for a return period of 50 years [52]. In addition, there are also some episodes of wind-induced wave regimes, usually associated with cyclonic lows that circulate close to the Iberian Peninsula (between 45 and 50 °N); however, the closest and potentially most harmful storms do not generate intense wave storms over the Cantabrian coast, as the strong winds blow offshore, which means they are not able to develop a long fetch. In addition, sea waves are also influenced by large-scale patterns of atmospheric variability, such as the North Atlantic Oscillation (NAO), East Atlantic (EA), and the West Europe Pressure Anomaly (WEPA) [34,36,48,53]. In summer, on the contrary, wind-induced wave regimes dominate under an expansion towards the NE of the Azores High.

Significant wave heights undergo a slight reduction when advances parallel to the coast, probably due to local effects of the bathymetry. While winter NW swells are the main erosive agent and sediment mobilizer, summer low-intensity waves from N and NE become the main agents of beach profile reconstruction. Sea level experiences a semidiurnal mesotidal regime, reaching values around 5 m during equinoctial spring tides. Atmospherically induced components in major storm surges reach positive anomalies around 50–70 cm [51].

Almost 1 million people live along the coastal municipalities, around 37% of the provincial total, and its average density of 761 inhabitants/km$^2$ is much higher than the

provincial one (331 inhabitants/km$^2$) [54]. Most of the population of the coastal areas is concentrated in a few cities, which usually correspond to the provincial capitals or constitute the most populated city of each province. Outside those cities, the landscape displays an extensive layout, with open nuclei, spread into single-family housings, linked to the rural and fishing activities. Such human pressure increases significantly in summer; despite not being part of the mass-tourism "sun and beach" circuits, typical of the Mediterranean coast, such activity has a long tradition and its contribution to both the GDP and the employment is remarkable. The volume of tourist activity (obviously not completely associated with the attractiveness of the coast as a whole) in 2014 contributed around 10% to both the GDP and employment [55]. In addition, its role as a booster of activity in other productive branches is evident.

This work has required the collection of complete records of oceanographic conditions and damage over a statistically representative period. Hydrodynamic data between 1996 and 2016 (7671 days), on an hourly scale, have been collected to identify the events that can be considered wave storms. On the other hand, post-disaster insured data corresponding to the "coastal battering" category have been compiled, for the same time span, to identify the episodes of damaging storms. Crossing both databases, it is possible to establish the relationships, i.e., statistical correlations, between the two parameters: the oceanographic forcing mechanisms and the direct damage consequences.

*2.2. Data Sources on Hydrodynamic Parameters of Wave Storms*

Direct measurements, from instrumental oceanographic data, were obtained from a tide gauge and a deep-water buoy, both operated by the Spanish Ports Authority (www.puertos.es, accessed on 3 May 2022) (Figure 1). After a data quality control analysis, events corresponding to wave storms were identified. They are defined as any temporal period during which the significant wave height (Hs) exceeded a threshold value, usually corresponding to a low percentage of exceedance in a dataset, in our case the 95th percentile (approximately 4.0 m) calculated from the dataset time series (1996–2016). No minimum storm duration was set, since in this way, we can analyze the impact of storm duration on the occurrence of damages. To ensure the independence of storm events, a sequence of wave heights above the threshold is considered a single storm if the interval between two storm peaks is separated by more than 30 h and the time lapse between the end of a previous storm and the beginning of a new one is longer than six hours; otherwise, they are regarded as the same storm.

For each one of the individualized wave storms events, other derived parameters were obtained: sea level height (SL: meters), average significant wave height (SWH, Hs: meters), peak wave period (Tp: seconds), duration (D: hours), and mean wave direction. Additionally, other parameters were calculated (Figure 3).

The Total water level (TWL: meters) is an index obtained by the combination of the main sea level components: the sum of astronomical tidal level, wind, and barometric induced set-up (storm surge), and the sea level variation due to the wave run-up (setup + swash). An estimation of the TWL can be achieved through the following equation

$$TWL = WL + R_2 \tag{1}$$

where WL is the measured sea level, encompassing the (atmospheric-) storm surge (SL$_{surge}$), the tide and the wave setup and run-up, and R2% is 2% exceedance value of wave run-up [56,57]. An estimation of the 2% exceedance value of run-up elevation can be derived from the following empirical formula [58]:

$$R_2 = 1.1 \left( 0.35\beta_f (H_0 L_0)^{\frac{1}{2}} + \frac{\left[ H_0 L_0 \left( 0.563\beta_f^2 + 0.004 \right) \right]^{\frac{1}{2}}}{2} \right) \tag{2}$$

where $0.35\beta_f(H_0L_0)^{\frac{1}{2}}$ is the wave setup component, $H_0$ the offshore wave height, and $L_0$ the offshore wavelength. Coast morphology along the northern coast of Spain is variable but, as we are not interested in TWL estimates for a specific section but rather an overall estimation, a theoretical representative beach slope ($\beta f$) of 0.05 was used.

| STORM REFERENCE NUMBER | DATE (first day) | | | SPI | Duration | TWL05_max | MEAN VALUES | | | | | | CONDITIONS OF MAXIMUM SEA LEVEL | | | | CONDITIONS OF MAXIMUM WAVE HEIGHT | | | | |
|---|---|---|---|---|---|---|---|---|---|---|---|---|---|---|---|---|---|---|---|---|---|
| | YY | MM | DD | m²h | h | m | Hs | Sea Level | TWL05 | Tp | Direction | Energy density | Sea Level | Hs | TWL05 | Tp | Hs | Sea Level | TWL05 | Tp | DirM |
| | | | | m²h | h | m | m | m | m | s | ° | J/m² | m | m | m | s | m | m | m | s | ° |
| 1 | 1996 | 1 | 1 | 319.71 | 12 | 5.18 | 5.13 | 2.63 | 4.02 | 12.1 | 292 | 391.63 | 3.65 | 5.7 | 5.18 | 12.4 | 5.9 | 3.15 | 4.77 | 13.2 | 291 |
| 2 | 1996 | 1 | 7 | 232.27 | 13 | 6.15 | 4.22 | 2.27 | 4.16 | 15.1 | 290 | 284.52 | 4.13 | 4.0 | 6.15 | 15.9 | 4.5 | 1.46 | 3.54 | 15.1 | 289 |
| 3 | 1996 | 1 | 12 | 1307.02 | 43 | 5.93 | 5.47 | 2.61 | 4.64 | 15.4 | 294 | 1601.05 | 3.87 | 4.9 | 5.93 | 14.1 | 6.4 | 3.59 | 5.70 | 16.4 | 293 |
| 4 | 1996 | 2 | 8 | 2606.29 | 63 | 6.46 | 6.22 | 2.55 | 4.16 | 11.9 | 307 | 3192.61 | 4.38 | 4.4 | 5.61 | 11.2 | 9.6 | 1.68 | 4.06 | 13.5 | 311 |
| 5 | 1996 | 2 | 10 | 1302.98 | 46 | 5.50 | 5.29 | 2.45 | 4.04 | 14.1 | 306 | 1596.1 | 3.77 | 5.5 | 5.50 | 15.1 | 6.0 | 3.63 | 5.44 | 14.9 | 303 |
| 6 | 1996 | 2 | 19 | 916.05 | 44 | 5.94 | 4.53 | 2.27 | 3.45 | 11.7 | 322 | 1122.13 | 4.73 | 4.5 | 5.94 | 12.4 | 5.5 | 4.18 | 5.60 | 12.4 | 320 |
| 7 | 1996 | 3 | 16 | 265.61 | 15 | 4.85 | 4.21 | 2.15 | 3.25 | 11.6 | 304 | 325.36 | 3.75 | 4.2 | 4.85 | 11.5 | 4.3 | 1.13 | 2.26 | 11.9 | 301 |
| 8 | 1996 | 5 | 2 | 241.13 | 12 | 5.43 | 4.48 | 2.55 | 3.71 | 10.3 | 298 | 295.38 | 4.33 | 4.2 | 5.43 | 10.3 | 4.8 | 1.15 | 2.39 | 10.4 | 297 |
| 9 | 1996 | 9 | 19 | 81.14 | 4 | 4.91 | 4.50 | 3.32 | 4.52 | 11.2 | 291 | 99.39 | 3.81 | 4.2 | 4.91 | 11.2 | 4.7 | 3.21 | 4.47 | 11.2 | 291 |
| 10 | 1996 | 10 | 5 | 238.72 | 14 | 4.18 | 4.13 | 2.31 | 3.61 | 14.2 | 317 | 292.42 | 2.91 | 4.1 | 4.18 | 13.7 | 4.2 | 1.72 | 3.08 | 14.5 | 316 |
| 11 | 1996 | 10 | 29 | 154.71 | 9 | 5.40 | 4.14 | 2.79 | 4.22 | 13.3 | 298 | 189.51 | 4.12 | 4.1 | 5.40 | 12.8 | 4.3 | 1.92 | 3.50 | 13.7 | 298 |
| 12 | 1996 | 11 | 5 | 442.48 | 21 | 4.55 | 4.58 | 2.35 | 3.70 | 12.7 | 306 | 542.02 | 3.15 | 4.9 | 4.55 | 13.0 | 4.9 | 2.79 | 4.16 | 13.2 | 306 |
| 13 | 1996 | 11 | 8 | 261.42 | 15 | 5.08 | 4.17 | 2.52 | 4.04 | 14.1 | 305 | 320.23 | 3.61 | 4.0 | 5.08 | 13.5 | 4.3 | 1.32 | 2.88 | 14.5 | 306 |
| 14 | 1996 | 11 | 20 | 1789.04 | 40 | 6.11 | 6.54 | 2.59 | 4.33 | 12.6 | 303 | 2191.51 | 4.02 | 8.0 | 6.08 | 13.5 | 8.4 | 2.67 | 4.86 | 13.7 | 303 |
| 15 | 1996 | 11 | 22 | 168.36 | 8 | 5.24 | 4.58 | 2.64 | 3.84 | 10.5 | 300 | 206.23 | 3.93 | 4.9 | 5.24 | 10.5 | 5.0 | 3.73 | 5.03 | 10.5 | 300 |
| 16 | 1996 | 11 | 28 | 186.61 | 11 | 5.07 | 4.12 | 2.51 | 3.56 | 9.4 | 304 | 228.59 | 4.03 | 4.1 | 5.07 | 9.4 | 4.2 | 1.91 | 2.98 | 9.4 | 302 |
| 17 | 1996 | 11 | 30 | 309.36 | 18 | 4.78 | 4.14 | 2.56 | 3.66 | 12.0 | 306 | 378.95 | 3.71 | 4.0 | 4.78 | 13.0 | 4.3 | 1.30 | 2.43 | 12.2 | 306 |

**Figure 3.** Extract from storm inventory. Data of some wave storms (in chronological order) out of the total of 346 in the database.

The storm surge ($SL_{surge}$) is calculated as the difference between the water level measured by the tide gauge (WL) and the predicted tide (deviation due to inverse barometer effect and the wind setup). The predicted tide, calculated from the main tidal constituents, was obtained with the R package OCE [59].

Among the various indices published in the scientific literature to quantify the energy generated by storm waves, we have chosen two very widespread and easy to calculate. Probably the best known is the Storm Power Index (SPI: m²h), defined by Dolan and Davis [60]. Several studies [61–64] have shown a strong correlation between SPI and damages recorded in coastal areas. This index combines the storm duration and the wave height according to the formulae:

$$SPI = H_s^2 td \tag{3}$$

being Hs the significant wave height (m) and td the storm duration, in hours.

We also calculated the mean wave energy density, whose values closely correlated with the wave energy flux index values proposed by Molina et al. [65] and used to quantitatively evaluate and classify storm waves in terms of their intensities. This index is calculated according to the formulae [59],

$$E = \frac{1}{16}\rho g H_{m0}^2 \tag{4}$$

where E is the mean wave energy density (J/m²), $H_{m0}$ is the significant wave height; $\rho$ the water density; and g the acceleration by gravity.

### 2.3. Data Sources on Indemnities for Damaging Storm Episodes

The effects of storm events have been assessed through the indemnities intended to cover the monetary losses, of insurance policyholders, not directly covered by insurance companies. Meyer et al. [66] distinguish between direct and indirect damage, and then subdivide them into tangible and intangible. Direct damage refers to the physical or structural impacts, while indirect ones are the subsequent or secondary results of the initial loss, such as the disruption of economic and social organization (usually over a longer temporal period and with wider spatial effects). Tangible (or quantifiable) damages

can be valuated monetarily (marketable goods and services), whereas intangible ones have no market value (environmental, health, and cultural impacts), and therefore have no assigned prize. Intangible and indirect effects are rarely considered because they are not easy to estimate nor integrate, despite the variety of valuation methods available [67]. Thus, direct tangible costs are usually regarded as a good indicator of the severity of a disastrous event [66]. In short, it is a monetary estimate of the cost invested in recovery as an objective indicator to "measure" the damage. Although this type of approach may be insufficient to carry out a complete damage assessment, the objectivity and precision of the data and the homogeneity of criteria in the consideration of indemnities, unchanged throughout the study period and on the whole study area, provide a high standard of objectivity and allow a consistent analysis of the relationships between the intensity of the storms and their impact. In addition, the insurance coverage ratio, close to 80% on average, varies slightly, but not significantly, between asset type and region, and places Spain among the top 10 countries in Europe in terms of insured assets [68,69]. However, above all, it is quite consistent over time. During the time span studied, there have been no socioeconomic changes relevant enough to influence insurance coverage. Therefore, it is a useful source of information for a comparative study to discriminate which characteristics of storms are associated with the highest damage. For all these reasons, this type of post-event economic damage study is a useful tool to justify future investments in adaptation strategies [70].

The Consorcio de Compensación de Seguros (CCS) [71], attached to the Ministry of Economy and Competitiveness, is the Spanish Public Reinsurance Entity, in charge of the coverage for extraordinary damage linked to "phenomena characterized by an absolute lack of regularity in their occurrence (both in frequency and intensity)", although not necessarily with the official classification of a catastrophic event". The damage data considered in this work are those due to "battering from coastal waters", that is impacts due to erosion and flooding, but excluding those due to wind or rain on inland areas, usually also associated with cyclonic storms. Compensation for damages due to this type of process corresponds in full to CCS, not being shared by any private insurance company [41].

It must be pointed out that only damage affecting insured assets, both privately and publicly owned, is considered. Therefore, a factor of uncertainty is the terms of the insurance contract concerning the guaranteed items: sometimes no assets are insured or only a fraction has insurance coverage [40,68,69,72].

The database provides complete temporal and spatial coverage for the period analyzed and for the study area. The data series used starts in 1996 when the CCS decided to distinguish this type of event from others caused by wind, rain, etc., and ends in 2016. Although an updated damage database is available up to 2020, the last years are still in the process of revision and are therefore considered provisional, so it has been decided to work only until 2016, corresponding to the data series already corrected by the CCS, and then considered definitive.

The information contained in the reports is not only detailed and exhaustive, but also disaggregated. Each record includes the date of occurrence, damage type, amount of compensation awarded, and specific affected locations (municipality and town). For data collection purposes, damage categories have been classified into three groups: real estate buildings (industrial, commercial, houses, offices, stores, etc.) vehicles, and infrastructure (ports, promenades, breakwater, pier and jetties, seawalls, roads, bridges, etc.). From all this information, a new database was created, grouping the data, initially on a daily scale, into damaging storm events (Figure 4). It should be stressed again that these damaging storms correspond to those with recognizable monetary impact and therefore represent a subset of the total number of wave storms that have occurred.

| DATE (first day) | | | DAMAGE BY PROVINCE AND TYPE (Indemnities €) | | | | | | | | |
|---|---|---|---|---|---|---|---|---|---|---|---|
| | | | Cantabria | | | Vizcaya | | | Guipúzcoa | | |
| YY | MM | DD | Buildings | Vehicles | Infrastructures | Buildings | Vehicles | Infrastructures | Buildings | Vehicles | Infrastructures |
| 2014 | 2 | 2 | 3,353,871.72 | 386,392.20 | | 900,171.86 | 30,273.00 | 3,051,248.55 | 8,102,051.97 | 86,790.18 | 1,907,314.22 |
| 2008 | 3 | 11 | 556,341.83 | 100,005.20 | | 286,133.00 | 34,448.47 | 1,941,643.96 | 8,684,541.94 | 237,933.46 | 3,731,961.35 |
| 1996 | 2 | 8 | 5164.87 | 4196.42 | | 3,473,659.30 | | | 222,593.16 | 285.90 | 24,232.81 |
| 2010 | 11 | 9 | 439,956.36 | 38,186.91 | | 1,706,929.19 | 13,025.00 | | 762,921.41 | 2857.50 | 497,270.28 |
| 2007 | 12 | 9 | 78,973.80 | 38,879.17 | | 201,449.66 | 5260.51 | 2,513,821.84 | 353,582.98 | 50,698.91 | |
| 2014 | 3 | 3 | 938,584.10 | 12,939.45 | | 63,814.90 | | 81,7961.18 | 291,048.44 | | 1,019,298.07 |
| 2007 | 3 | 19 | 84,517.16 | 8891.70 | | 3240.12 | | | 1,004,343.38 | 31,408.03 | |
| 2014 | 1 | 28 | 77,649.22 | 13,669.70 | | 5816.24 | | 470,397.43 | 735.86 | 3120.63 | 557,705.30 |
| 2014 | 1 | 4 | 678.90 | | | 42,724.00 | | 557,900.95 | 125,413.14 | 1241.86 | 136,883.46 |
| 1998 | 1 | 2 | | | | 297,760.91 | | | 377,065.47 | 97,973.52 | |

**Figure 4.** Extract from the inventory of damaging storms. Indemnity data for the 10 major damaging wave storms.

*2.4. Procedure*

Linking recorded damaging storm episodes to the hydrodynamic-forcing parameters ultimately responsible for them allows the identification of the most important drivers for coastal damage and the establishment of tentative storm thresholds, at a regional scale. The most relevant parameters (sea level, duration, Hs, TWL, or combinations of them) have been tested to determine the most suitable hydrodynamic conditions to explain the magnitude of the monetary impacts.

Given the availability of damage data by type of asset affected, it has been possible to carry out a more detailed analysis and look for specific thresholds for each type of damage (buildings, infrastructures, and vehicles). In addition, some singular episodes, capable of producing numerous and severe damage simultaneously, and affecting large areas have been identified.

Further validation of the thresholds requires comparison with storm events with little or no damage. In this context, each storm event was classified into three categories according to the amount and distribution of monetary losses:

- No damage: storm event in which no damage has occurred.
- Little damage and of a local character, that is, affecting a few locations
- High-impact storms, which, due to their exceptional nature have caused significant damage, in terms of quantity, and were distributed over multiple areas during the storm life cycle, affecting all provinces and most of their municipalities (multisite events).

As an indicator of the degree of the exceptionality of major damage storm events, a percentile-based approach has been proposed. The calculation of the empirical values has been based on the entire storm event database reported for the 21-year study period, i.e., from the oceanographic data of the identified storm events.

Finally, one of the most common procedures, widely proposed in the literature, to find relationships between rainfall and landslides, has been adapted. The aim is to establish a quantitative correlation between rainfall intensity and duration: rainfall events leading to landslides are plotted in Cartesian, semilogarithmic or logarithmic, coordinates, and then the best-fit line is obtained. In a log(I) vs. log(D), a power law curve is assumed, so the I-D equation has the general form $I = \alpha D{-}\beta$, where I is the intensity, D is the duration, $\alpha$ is the intercept, and $\beta$ defines the slope of the power function [73,74]. By applying the same procedure to the occurrence of the major damaging wave storms, it is possible to obtain a tentative empirical threshold Intensity–Duration function, where I is the storm intensity (energy density/duration: $J/m^2/hour$) and D its total duration (hour). The threshold, in this work, is represented by the equation resulting from shifting the obtained best-fit line to match the lower boundary of the empirical data. That is, the function obtained represents

the lower-bound line and describes the critical intensity-duration conditions above which extreme damage is triggered.

## 3. Results

### 3.1. Spatial and Temporal Distribution of Storms

Storm events occur almost every extended winter season (October to March); however, despite the high-intensity wave regime of the study area, only a few events are regarded as having caused any damage at some point on the coastline. Over the 21-year period, 346 wave storms were identified but only 68 (~20%) correspond to damaging storm episodes. The direct economic losses (indemnity granted) were EUR 51,572,106.43 (Figure 5).

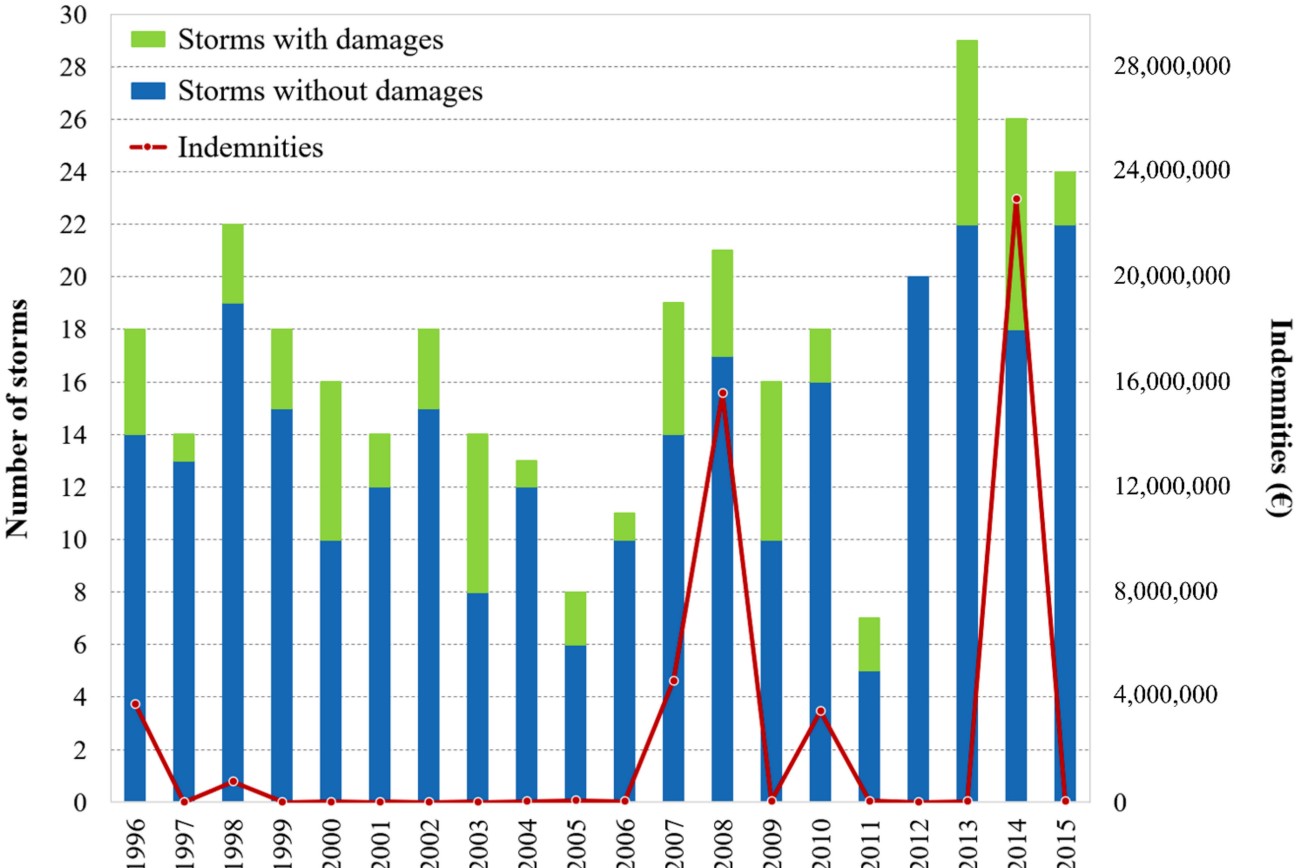

**Figure 5.** Interannual distribution of storms: number of wave storms and damaging storms, and amount of indemnities.

In terms of the type of damage, most of the claims concern real estate property, especially commercial premises. However, it is observed that indemnities increase exponentially when engineering infrastructure is affected (Figure 6). This is not only due to the high costs of such infrastructure, but because damage to such assets is indicative of exceptionally intense storms causing high damage to all categories of assets and along the entire length of the coastline.

As displayed in Figure 6, most of the storms recorded in our database caused slight damage. These damages correspond mainly to vehicles parked in the vicinity of the urban promenades or homes or businesses near those urban promenades. Only a few storms generate most of the damage; those storms can be visualized by the jump in the bars of the figure, which correspond to the interval between EUR 100,000 and 1,000,000 (remember that the scale of the vertical axis is not linear but logarithmic.

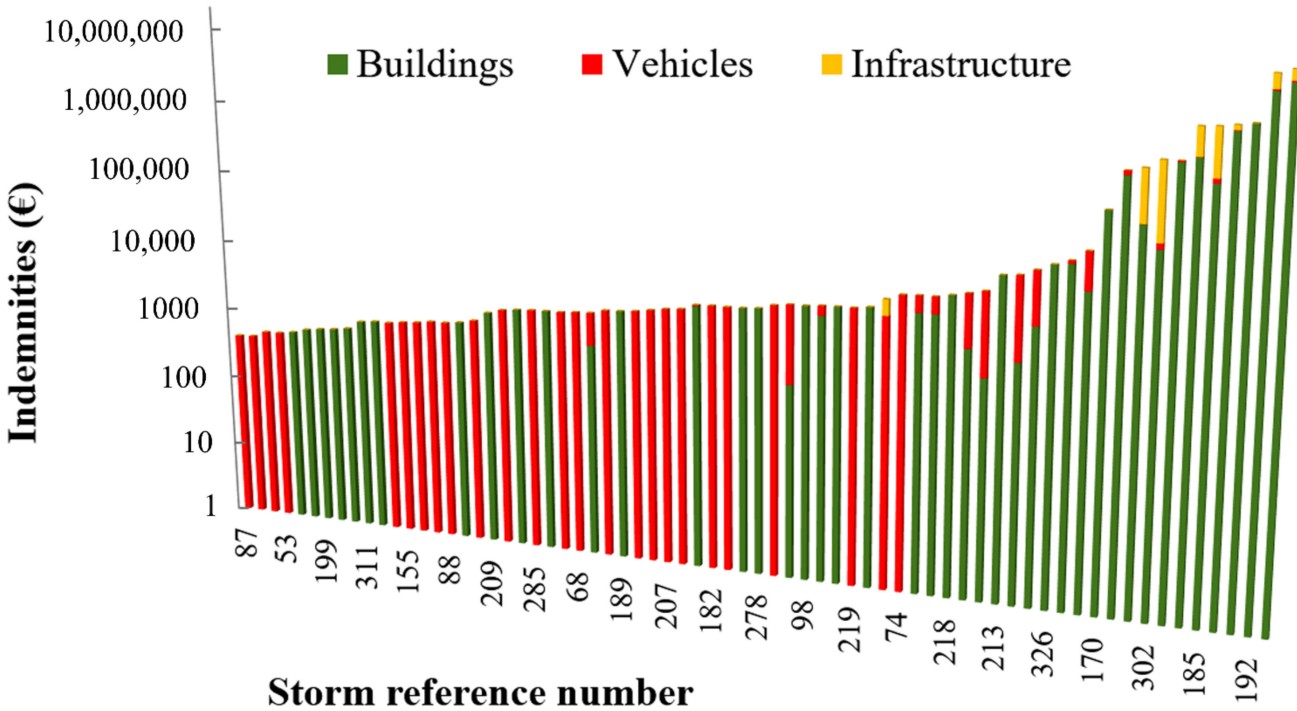

**Figure 6.** Classification of compensation amounts in the different categories of damage.

However, to delimit that threshold more precisely, we submitted the indemnities corresponding to all storms to an optimization classification method, the Jenks natural breaks method. This is a data clustering method, which seeks the best arrangement of a series of values into different classes. This is done by seeking to minimize each class's average deviation from the class mean, while maximizing each class's deviation from the means of the other classes. We repeated this procedure several times with a different number of classes, but the membership of the group of most damaging storms did not change. The value of EUR 500,000 serves as a convenient and easy-remembered threshold.

Results show an unequal distribution of damages, both from the geographical point of view and the temporal frequency within the studied period. Most of the monetary losses (98.6%) correspond to 10 major storm events, those that have caused significant, widespread, and simultaneous damage along the entire coastline studied, and 64% to only 2 storm events (10–13 March 2008; 31 January–3 February 2014) (Figure 7). Particularly outstanding was the 2013–2014 winter season, the most energetic on record, due to an accumulation of 18 storm events, almost consecutively. In addition to the huge direct economic losses (accounting for more than 50% of the total), there were widespread and large-scale erosion and flooding processes along the entire coastline of Western Europe [13,14,26,39,75,76]. The remaining storms show much lower damage figures (less than 500,000 EUR/episode) and a much more limited spatial distribution, affecting, in some cases, only a single coastal locality.

In addition to an evident temporal variability, the amount and type of monetary compensation show a contrasting spatial distribution (Figure 8). If the total amount of economic losses is discretized by municipality, it becomes clear that most of the damage is concentrated in a few municipalities, corresponding to the largest urban areas or harbors.

### 3.2. Regional Empirical Thresholds

The oceanographic parameters that play an important role in the impact of a storm, and which are often considered in the literature to establish thresholds, are wave height, sea level, TWL, SPI, and storm duration. For each of these parameters, the average value, corresponding to the whole storm event, and the maximum value reached at some time during the episode have been applied.

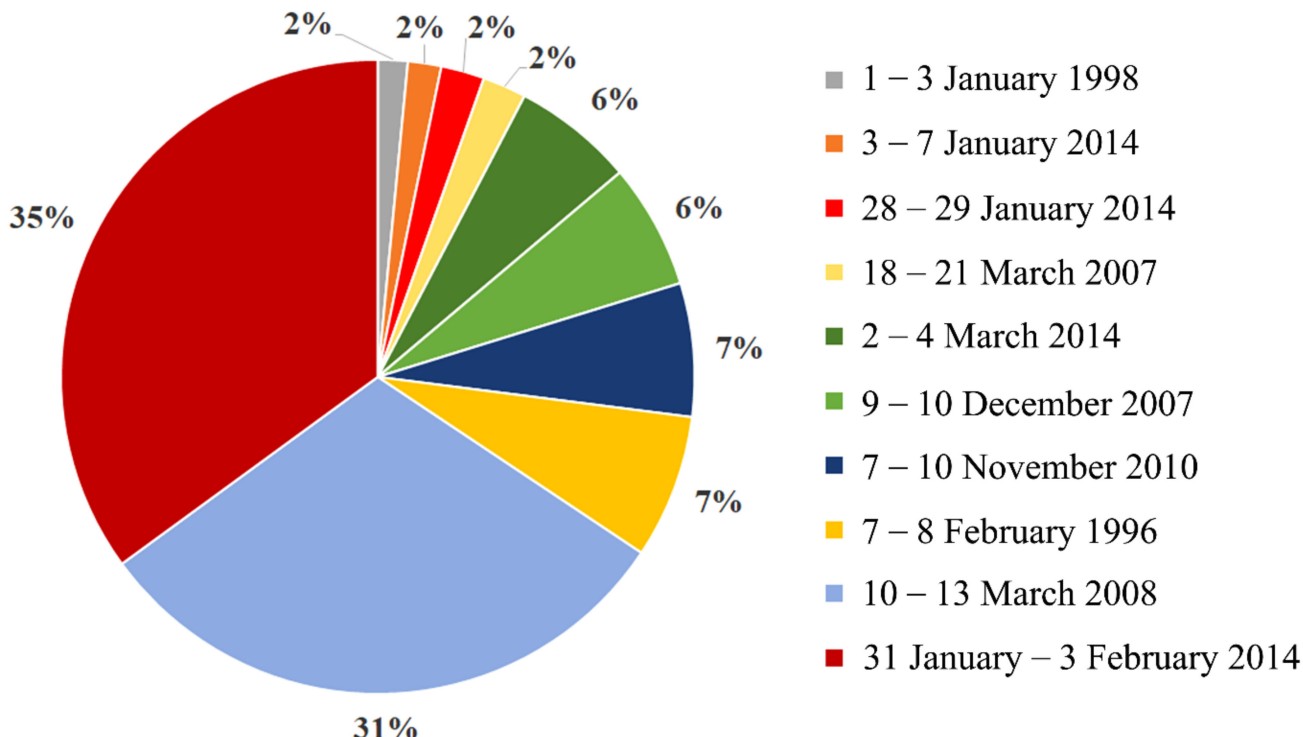

**Figure 7.** Percentage of damage and dates of the 10 most relevant storm damage events during the time span analyzed.

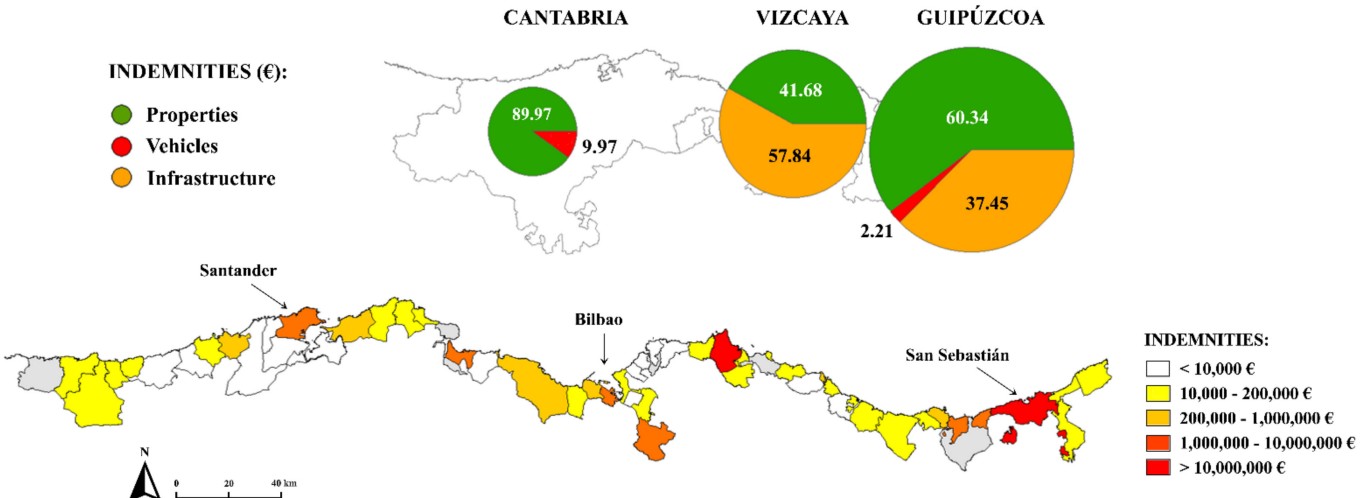

**Figure 8.** Spatial distribution of indemnities.

Figure 9 shows the values reached by these hydrodynamic parameters for the 68 damaging storms. From these results, no conclusive and meaningful relationship for the establishment of thresholds can be obtained. Only the direction (Figure 9e) shows a significant, but also obvious and expected result: the damage occurs exclusively with NW storms (between 290 and 340°) in which, in addition to their intensity, the large fetch (more than 1000 km) must be added; the swell from the West, also with a large fetch, suffers a slight reduction in wave height when it advances parallel to the Cantabrian coast.

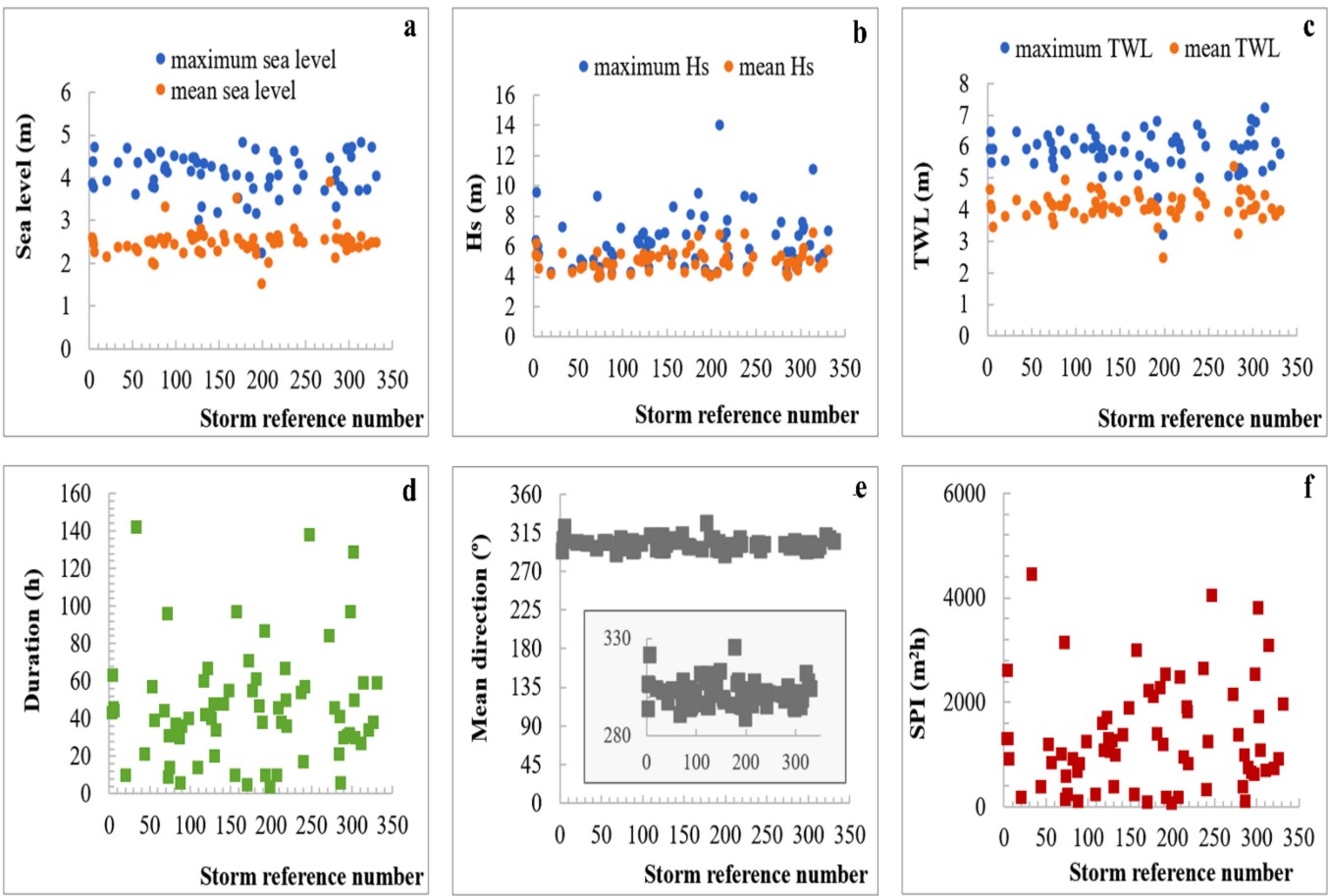

**Figure 9.** Values of the different hydrodynamic parameters for each damaging storm episode, in chronological order: (**a**) Sea level; (**b**) Wave height; (**c**) Total water level; (**d**) Duration (green); (**e**) Mean direction (grey); (**f**) Storm Power Index (red).

However, by analyzing the amount of damage and type of damage produced, it is possible to identify some more significant relationships (Figure 10). The maximum wave height (Figure 10b) does not seem to be a very relevant indicator of damage to buildings and vehicles; however, only heights higher than 6.5 m can cause significant damage to the resilient coastal infrastructure. In addition, a storm duration of more than 24 h, and in most cases 48 h, is needed (Figure 10e). The integration of both parameters in SPI gives a minimum value of 1700 m²h (Figure 10d). A similar situation occurs with sea level, only above 4 m does damage to coastal infrastructure occur (Figure 10a). The Total Water Level (TWL) marks the flood level of the coast, with maximums when spring high tide coincides with high storm surge and waves reach supratidal areas, usually with human settlement. When the TWL is below 3 m, no damage is observed; when TWL reaches values above 5 m, damage is mainly caused by flooding of built-up areas (houses, commercial premises, and offices) and especially vehicles. Damage to infrastructure only occurs in the strongest events (TWL > 6 m) (Figure 10c). In these conditions, in addition to the over-washing of the hinterland, there is the wave impact effect, produced by repeated wave breaking, which is associated with high run-up values. In summary, the occurrence of over-wash and coastal flooding is highly dependent on sea level and significant wave heights offshore, conditions that can occur even during low-energy storm events. Therefore, high energy waves produce dramatic coastal erosion and damage to structures, but only if they coincide with high tides can they also affect the hinterland.

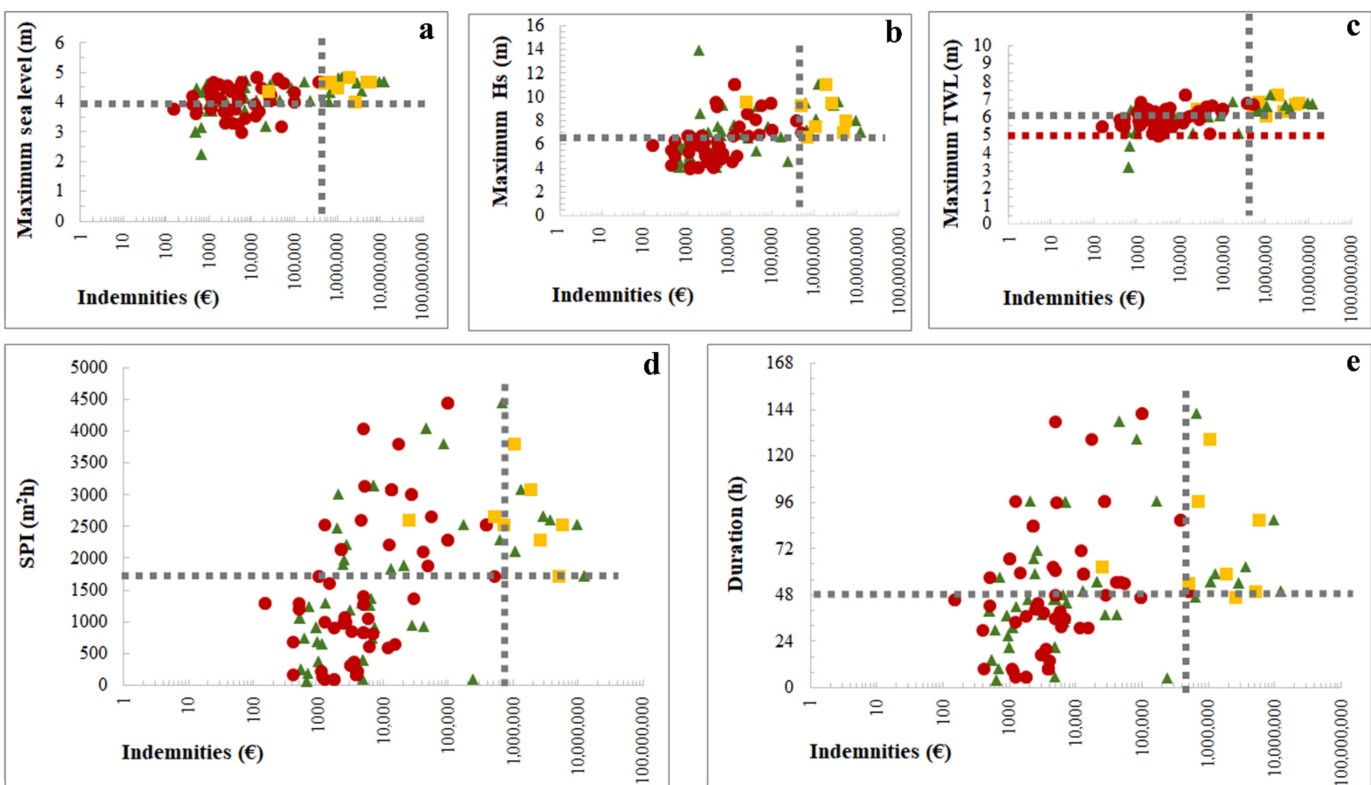

**Figure 10.** Thresholds for various hydrodynamic indicators based on the amount and type of damage on the eastern Cantabrian coast. Buildings: green triangles; Vehicles: red circles; Infrastructures: yellow squares. Dotted grey lines: thresholds for infrastructure. Dotted red line: threshold for vehicles. (**a**) Maximum sea level; (**b**) Maximum wave height; (**c**) Maximum total water level; (**d**) Storm Power Index; (**e**) Duration.

The same thresholds are valid for discriminating conditions that caused severe damage (dashed vertical lines in Figure 10). Damage to infrastructure requires more energetic conditions, related to strong waves; in these cases, the damage to buildings and vehicles is also greater. In other words, major impacts are correlated with storms that affect infrastructure, even if they do not account for most of the damage. In conclusion, there is no clearly defined threshold to explain the occurrence of damage, but it is possible when it concerns relevant damage.

In this context, it is interesting to extract the critical thresholds corresponding to the 10 largest storms (Figure 7) in terms of magnitude and spatial extent of damage. A validation is carried out by comparing the values of the different parameters needed to produce severe damage (EUR > 0.5 million) and widespread effects (affecting most stretches of the coast), with storm events with no or little damage.

The oceanographic characteristics associated with the 10 major episodes identified cover a broad spectrum (Figure 11), in terms of maximum sea level (4.02–4.84 m), wave height (6.7–11.1 m), and TWL (6.06–7.25 m), as well as energy (1725.97–4460.85 m$^2$h) and duration (47–142 h). Confirming the results obtained in Figure 10, the threshold for significant and widespread economic losses would be set as duration >48 h, energy >1700 m$^2$ h, maximum wave height >6.5 m approaching from the NW quadrant, TWL >6 m, and during spring tides >4 m. A sea level of 4 m was exceeded by only 4%, and the wave height (6.5 m) and TWL (6 m) represent the 99th percentile. Durations longer than 48 h and an SPI of 1700 are values that correspond to the 85th and 90th percentile, respectively, considering storm events between 1996 and 2016. This analysis shows the extraordinary nature of these storms, which are responsible for most of the damage. As can be seen, most of the storm episodes without damage show much lower figures.

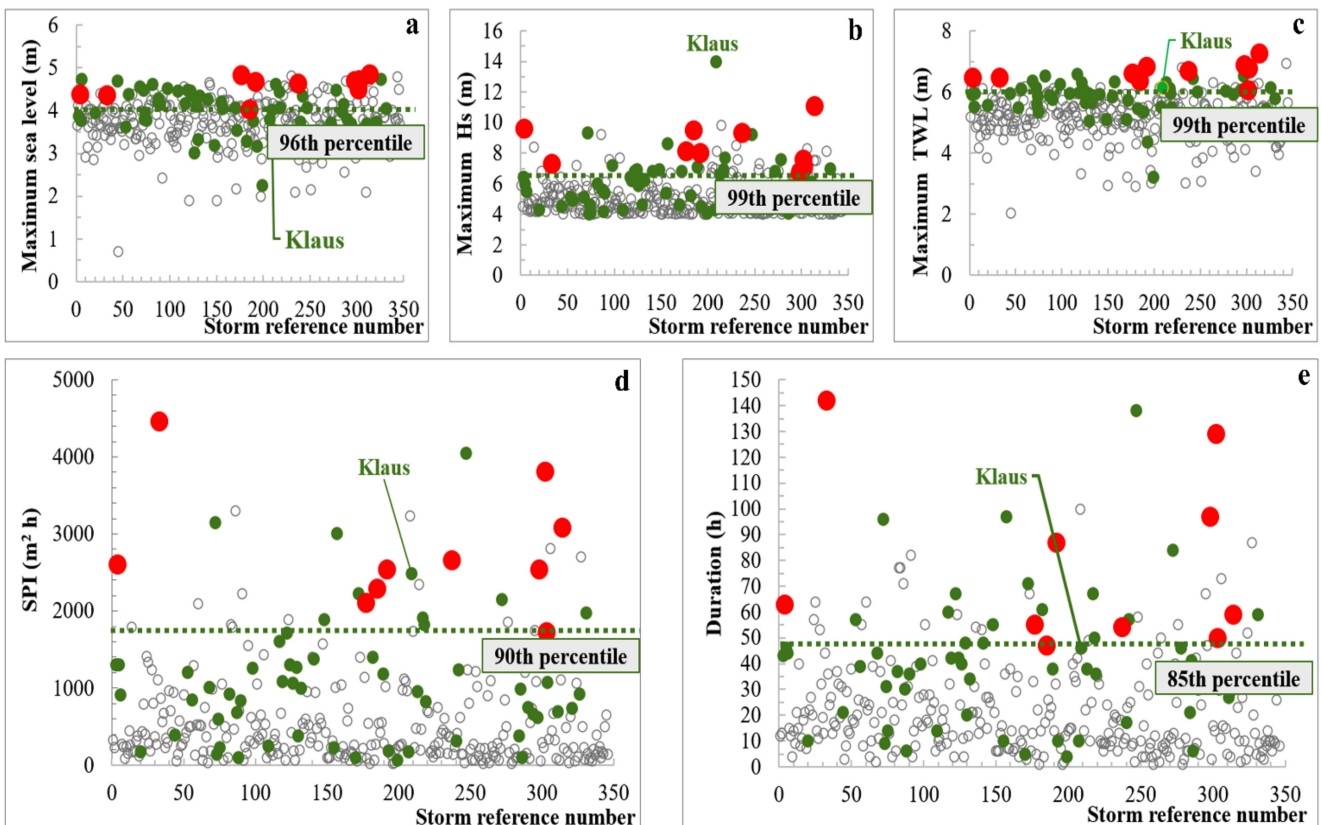

**Figure 11.** Thresholds of different hydrodynamic indicators corresponding to the top 10 damaging storms The lines show the empirical percentiles obtained for the 21-year storm database. No damage: hollow grey circles; Little/local damage: grey circles; High-impact storms: red circles. (**a**) Maximum sea level; (**b**) Maximum wave height; (**c**) Maximum total water level; (**d**) Storm Power Index; (**e**) Duration.

The specific characteristics of the different types of coastline, both in terms of the quantity and economic value of the elements exposed to risk and their vulnerability, and their physical factors (orientation and morphology, which control how waves and tides affect the coastline) [70], do not allow the definition of universal thresholds. Therefore, the values obtained here are thresholds with a very regional character, and hence, it does not make sense to extrapolate them from one area to another. In addition, it is difficult to compare the results obtained by other authors, due to different starting data and methods. However, it may be useful to make comparisons with the most representative thresholds obtained for similar or nearby geographical areas, or even from similar damage data assessments; in this case, from damage data regarding human occupation. In the latter, what is relevant is not the comparison between threshold values, but the consideration of the parameters that have been found to be the most representative, not always coinciding with those necessary to produce, for example, erosion in beach morphologies.

In the Basque Country, corresponding to the easternmost coast of our study area, and where most of the damage accumulates, Gaztelumendi et al. [77] use the same sources of information as in this work, but for a shorter period (10 years). Their results show that most of the adverse events occurred with significant wave heights surpassing 7 m, and the most harmful coinciding with tide levels over 4.5 m. In our case, half of the events with major damage exceeded these figures. For the same area, Euskalmet (Basque Meteorological Agency) establishes a coastal and maritime "traffic light" system (warning/alert/alarm) for impacts coming from the direct push of waves and flooding, by means of the so-called "overtopping index". This index is understood as the height that the water can reach due to the combined effect of tide and waves on a coastline characterized by average

conditions [78]. This index (I), broadly comparable to the TWL used in this work, ranges from I > 5.75 for initiating the warning, a figure like the one obtained here, to I > 7.25 m, for the alarm situation. It is reasonable to think that the differences found may be due, in part, to the much shorter length of coastline considered and, above all, to the bias derived from being the stretch most affected by damaging storms (Figure 6).

In the Ancão Peninsula, on the Atlantic coast of southern Portugal, from historically documented major storms, responsible for similar damage to those considered in this work, and using the same hydrodynamic factors, the following thresholds for individual storms have been obtained: maximum Hs of 4.7 m and 6 m, depending on wave approach direction, and a duration of 2 days [16,20]. Although the wave height values are slightly lower than those obtained in this work, as expected due to a clear difference in geographical location, there is a clear coincidence in the importance given to a minimum duration to produce significant damage.

In an area close to the previous one, del Rio et al. [21] provide values for the minimum hydrodynamic conditions needed to produce direct damage, from newspaper reports. They refer to "those events for which there is a written record of their destructive effects" on coastal infrastructure or human occupation. Those storms should be considered as being the most damaging storms and therefore comparable to the 10 major storms used in this work. They also found a maximum wave height higher than 7 m in some specific local sectors, but >4 m, during spring tide conditions, and storm duration of 30 h or more, when considering the whole Spanish coast of the Gulf of Cádiz, in line with the spatial scale of this work. It is obvious that the wave height values obtained have nothing to do with those found on the eastern Cantabrian coast, but they do show that although "several events with relatively short durations have caused reported destruction", as is also the case in the Cantabrian (Figures 7 and 8), it is the long-lasting events that generate significant damage.

Both the above results and those obtained in this work indicate that none of the main hydrodynamic parameters, considered independently, show a good correlation with their effects and that, therefore, a conjunction of conditions is required, conditions whose probability of occurrence is low, as shown in the case of SWH and SL (Figure 12).

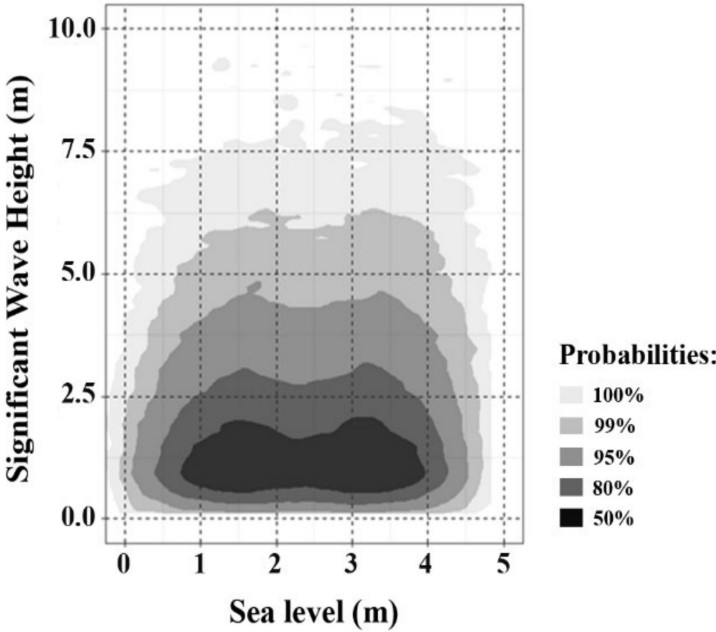

**Figure 12.** Joint probabilities of the significant wave height and sea level in the Eastern Cantabrian Sea from 1996 to 2016.

All the findings seem to suggest that the severe damage on the eastern coast of Cantabria is linked to intense long-duration storms. Thereafter, an approach is proposed

that combines the storm energy, which depends mainly on the wave height, with its duration. That is, occurrence of damage as a function of intensity and duration. Both parameters are plotted in a log(I) vs. log(D) graph and the distribution fitted to the power law equation. The calculated threshold relationship is fitted to the lower boundary of the storm conditions leading to damage. The obtained I-D function therefore represents the lower limit, where the best-fitted line has been shifted downward until it low-bound all ten major events, and then it refers to extreme conditions that enable the simultaneous occurrence of severe damage (Figure 13). The resulting expression represents the minimum threshold below which no extreme damaging events were recorded. The interpretation of this figure is as follows: theoretically, a vertical slope of the fitted line would indicate that, in the occurrence of a severe damage event, the decisive parameter is the duration of the wave storm, while the intensity would be irrelevant. Conversely, a horizontal fit would show that the duration of the storm does not contribute to the magnitude of the damage. In this case, the resulting expression for this function, $I = 248.7 \, D^{-0.453}$, shows the usual descending trend in which the minimum amount of storm intensity required to produce damage decreases with increasing storm duration, as reported in the literature for other kinds of processes. The intensity is crucial to produce the damage associated with the wave collision with structures and the duration to allow the coincidence, at some point, of high wave energy with high tide conditions. Although storm duration is not necessarily "critical in determining the occurrence of coastal infrastructure damage" [21], a long duration is decisive, in order to cover the maximum possible full tidal cycle and thus have a better chance of reaching a sufficient TWL to overflow.

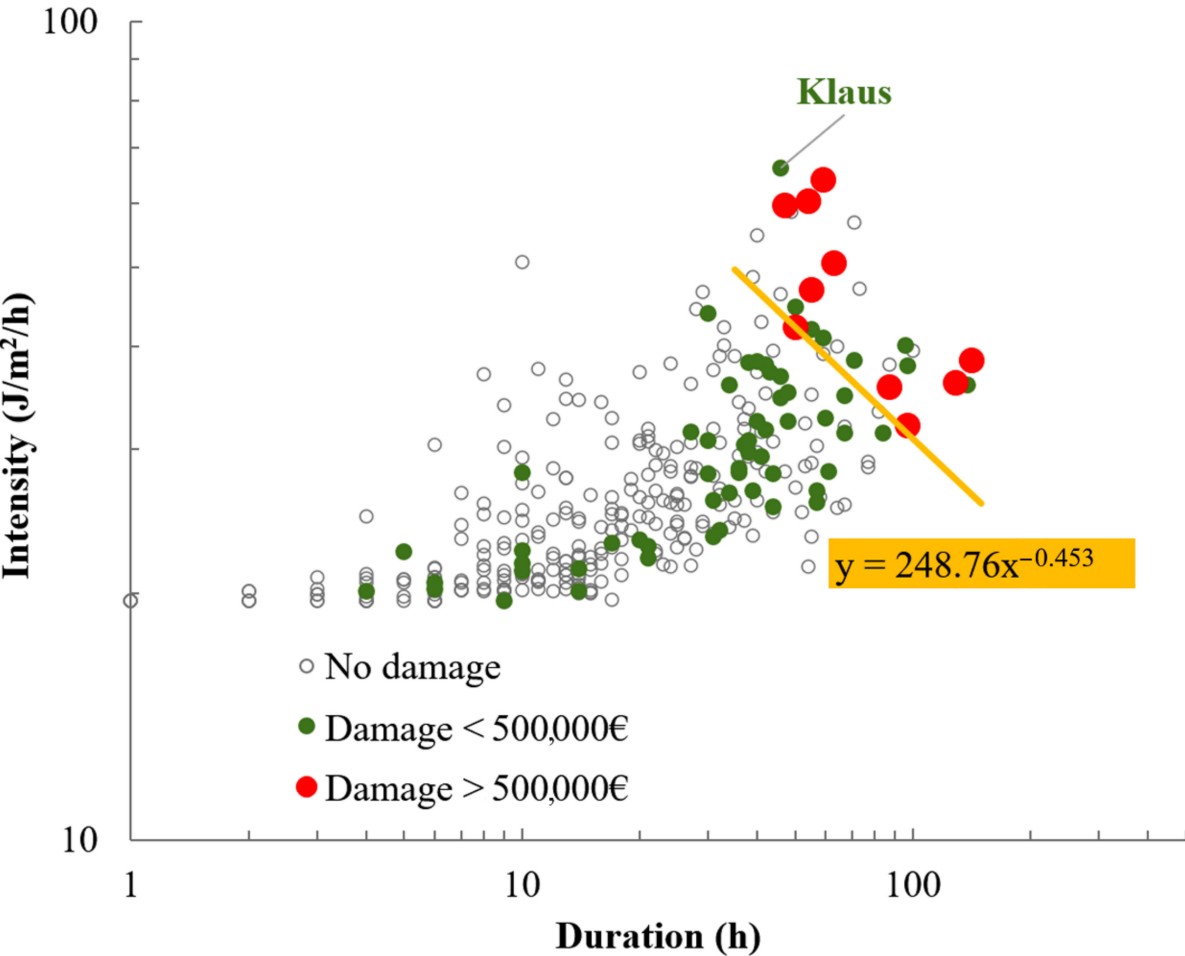

**Figure 13.** Intensity–Duration Log-Log plot. I-D function shows that increasing the duration decreases the minimum intensity needed to trigger significant economic impacts.

Obviously, damage may occur under less extreme storm conditions than the threshold values obtained here. Again, the threshold does not refer to the minimum conditions required to produce economic damage, but to the critical values and the minimum hydrodynamic conditions capable of producing significant and multiple damage to coastal human goods, and with a wide spatial distribution. The results also show that storms of extreme intensity or high sea levels do not always produce severe effects and, on the contrary, much less powerful episodes can generate significant damage. In other words, exceeding the critical threshold does not invariably mean that severe damage will occur, but there is a high probability that damage will occur. The established thresholds are affected by some uncertainty and may lead to "false positives", i.e., they may predict damage that does not occur. Conversely, there is always a (small) probability of damage occurring with storm conditions below the threshold values.

To illustrate those statements, we can compare the meteorological characteristics and the damage associated with two major storms that crossed the Bay of Biscay during the study period Klaus (23–24 January 2009) and a storm dated 7–8 February 1996 (Figure 14 and Table 1). Despite displaying comparable atmospheric conditions, they were responsible for substantially different contrasted losses on the eastern Cantabrian coast (EUR 1937.51 and EUR 3730,132.46, respectively). Such disagreement is particularly relevant for Klaus, considered one of the strongest storms affecting the Iberian Peninsula in the recent decades [23], but at the same time, one of those storms that did not fit the proposed thresholds, which highlights the need to consider more broadly the decisive parameters in the occurrence of serious damage.

**Table 1.** Meteorological characteristics of the two major storms that crossed the Bay of Biscay during the study period: Klaus (23–24 January 2009) and a storm dated 7–8 February 1996.

| Parameter | 7–8 February 1996 | 23–24 January 2009 (Klaus) |
|---|---|---|
| Minimum sea level pressure (Igueldo observatory) | 1000.2 (09 UTC, 7 February 1996) | 985.2 (00 UTC) |
| Maximum wind speed (Igueldo observatory) | 320°/139 km/h (20 UTC) | 270°/126 km/h (07 UTC) |
| | WD/SWH/TP/SL/ TWL/UTC [1] | WD/SWH/TP/SL/ TWL/UTC [1] |
| At the maximum SWH | 310/9.6/13.7/2.37/ 4.75/01 | 294/14.0/15.7/1.58/ 4.79/09 |
| At the maximum TWL | 313/8.9/13.7/4.26/ 6.46/05 | 298/10.0/14.9/3.69/ 6.14/14 |

[1] WD stands for wave direction (degrees); SWH for Significant Wave Height (meters); TP for Wave Period (seconds); SL for Sea Level (meters); TWL for Total Water Level (meters) and UTC for UTC hour.

Both storms followed a similar track, from the North Atlantic to Western Europe, crossing the Gulf of Biscay; Klaus being the fastest. In both events, after the cold front, strong northwesterly winds battered the coast of Northern Spain with winds exceeding 30 m/s (thus, a hurricane force). Although Klaus was deeper, its winds were not so fast and took a more westerly component, producing a SWH of 14 m (and an instantaneous maximum of 20 m), the absolute record of the period of analysis. Conversely, the low of February 1996 was shallower and only recorded a SWH of 9.6 m. However, the wave storm persisted for enough time to make coincident high waves with a high sea level (4.26 m), producing at 05 UTC a Total Water level of 6.46 m, above the corresponding maximum TWL of Klaus, which was only 6.14 m, due to its coincidence with a low sea level (3.69 m). Such high TWL caused flooding of large coastal areas and severe economic losses. In summary, spring tidal levels coupled with positive surge events enhance the probability of coastal flooding by extreme water levels; by comparison, a surge peak at low astronomical tide is less likely to produce flooding, or, if it does, of much lesser magnitude. Therefore, this

comparison shows the reduced role of atmospheric forcing on sea level (pressure and wind set up). The astronomical tide has a relevant influence on coastal flooding events, while storm surges play a secondary role.

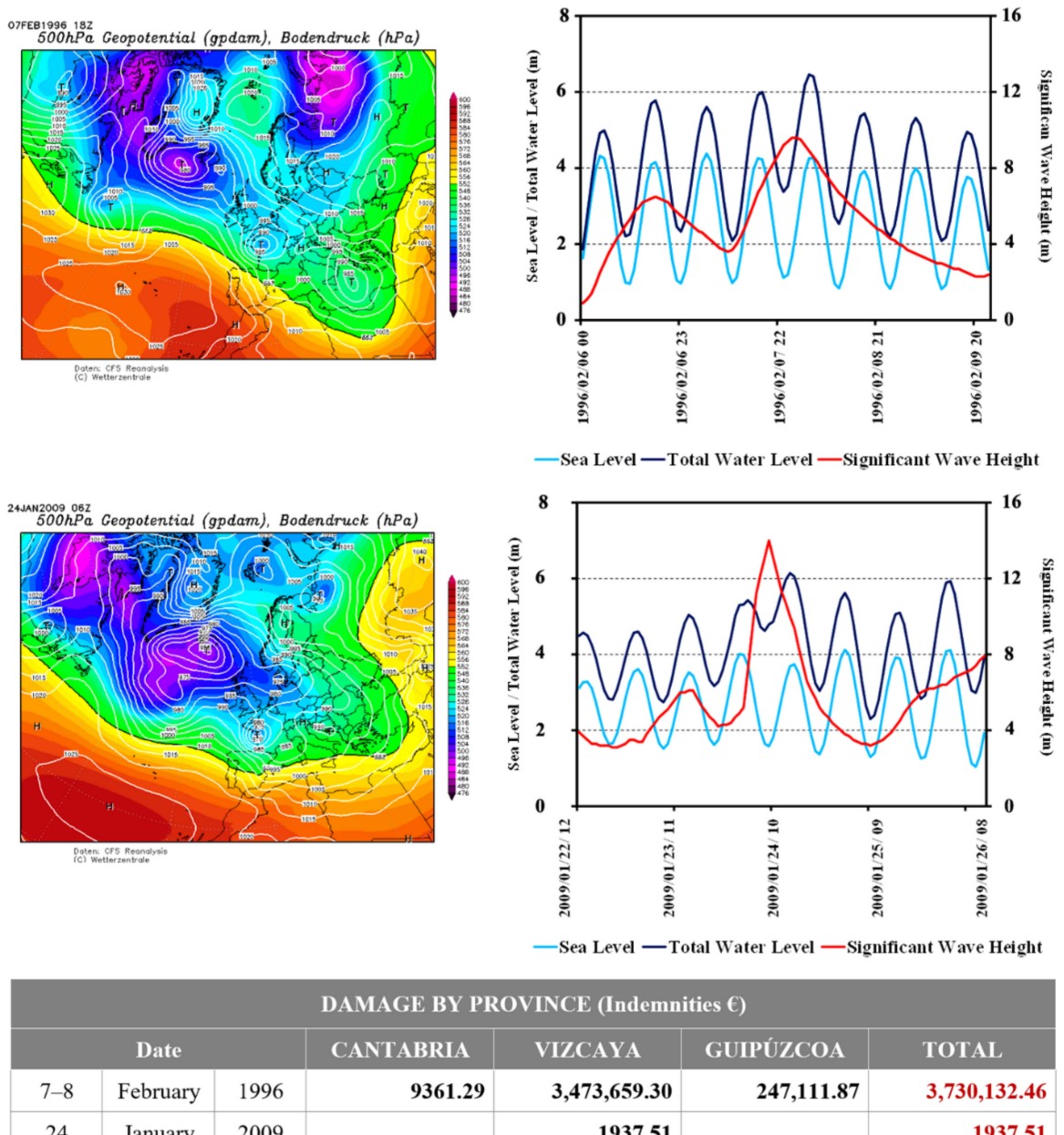

**Figure 14.** Weather charts (https://www.wetterzentrale.de/, accessed on 3 May 2022), oceanographic conditions, and amount of indemnities during two selected storms [79].

In summary, the large bulk of reported damage can be attributed to a limited number of storms combining high, but not necessarily exceptional, wave height and high spring tides. This is in line with results obtained by other authors, both in the study area [24,27,49], as well as in others in the Gulf of Leon [80].

Another factor, not specifically analyzed in this work, which should be considered, is the number of successive high-energy storms, as the impact of clusters of storms is known to play a significant role, at least in the morphological responses of the beaches. In such cases, the cumulative effects of successive storms are combined with the lack of beach recovery between them, resulting in a retreat of the shoreline; thresholds are

then expected to decrease [16,81]. This type of situation occurred during the winter of 2013–2014, with nine consecutive storms. Data from the European Centre for Medium-Range Weather Forecasts (ECMWF) catalogue indicate that while this season was not particularly exceptional in terms of the intensity of the storms, it was exceptional in terms of their recurrence [74,75]. In fact, the temporal clustering of the 2013–2014 winter storms is responsible for an unprecedented erosion of the main beach–dune systems with, in some cases, irreversible regression, at least in the medium term [25,28]. That winter, the investment for coastal recovery along the entire Galician–Cantabrian coast was about EUR 70 million [24]; about half of it refers to direct and tangible damage as considered in this paper. However, this perception that storms of moderate intensity, but close in time, can cause greater damage may be valid for infrastructure, but does not seem to play such a relevant role in the case of damage to buildings or vehicles, and even, as pointed out by Gaztelumendi et al. [77], economic losses in previously damaged areas tend to be less significant.

Despite the uncertainties and subsequent limitations, the results obtained constitute an important contribution to the understanding of storm conditions responsible for major coastal losses and can be very useful for improving hazard assessments. These are reliable thresholds that could be incorporated into operational early warning systems, reducing the number of false or failed alarms. An updated database, with a larger number of new events, would allow for the validation of the thresholds obtained, or their refinement, and would also improve their statistical significance.

Finally, the values obtained need to be continuously updated due to changes in human coastal occupation over time and the temporal variability of sea level and wave climate (frequency, intensity, and duration of extreme storm events) linked to climate change. Future projections regarding variations in the frequency and/or intensity of these extreme episodes are still uncertain, but it is clear these extreme wave events are likely to continue in the future and should be planned and prepared for.

## 4. Conclusions

Over the 21-year period, 346 wave storms were identified of which only 68 are damaging storms. The results show an uneven distribution of damage, both geographically and in terms of temporal frequency within the studied period. Most of the monetary losses (98.6%) correspond to 10 major storm events, which provoked the most severe, widespread, and simultaneous damage along the entire coastline. Guipúzcoa, at the eastern end of the coast, was by far the most affected province.

No conclusive and significant relationship between oceanographic values and total damage was found. However, when disaggregated by the amount of damage and type of asset affected (buildings, infrastructures, or vehicles), it is possible to identify some more meaningful relationships.

Although, with limitations, reliable thresholds were obtained for the major damaging storms. Results show that the critical conditions leading to the triggering of severe coastal damage over large areas are the result of a combination of parameters: a duration >48 h, with a maximum wave height >6.5 m approaching from the NW quadrant; TWL > 6 m, and during spring tides >4 m. The exceptional nature of these large damaging storms is evidenced by the fact that the values of most of their hydrodynamic parameters exceed the 95th percentile. These values are therefore much higher than those for storm events with little or no damage. An intensity–duration threshold defined by the following function was also obtained: $I = 248.7 \, D^{-0.453}$. The major impacts are produced by high tides combined with concurrent extreme waves, i.e., they play a more important role than surges during high seas (large waves). This combination is more likely to occur during long-lasting storms.

The long period covered by this study provides a representative picture of the oceanographic conditions that trigger major storm events. Naturally, there are other events that do not comply exactly with the patterns described here, but the discussed events undoubtedly constitute the most significant ones in terms of quantity and diversity of damage, as well

as spatial distribution. Therefore, these results constitute a significant improvement in the knowledge of storm damage conditions, which can be used, along with weather forecasting, for the early warning system of storms in the region. In addition, knowing in advance the type and magnitude of expected damage makes it easier to implement preventive mitigation measures.

**Author Contributions:** Conceptualization, V.R., C.G. and D.R.; methodology, V.R.; formal analysis, C.G. and V.R.; writing—original draft preparation, V.R.; writing—review and editing, C.G. and D.R. All authors have read and agreed to the published version of the manuscript.

**Funding:** This research was funded by the research project 17.JU11.64661 "Climatología Histórica de temporales en el área cantábrica (1851–2015)", funded by SODERCAN S.A. (Sociedad para el Desarrollo Regional de Cantabria) and Programa Operativo FEDER, and the support of Biodiversity Foundation of the Ministry for Ecological Transition of Spain.

**Institutional Review Board Statement:** Not applicable.

**Informed Consent Statement:** Not applicable.

**Data Availability Statement:** Not applicable.

**Acknowledgments:** The authors are deeply grateful to Alfonso Nájera and Francisco Espejo, from the *Consorcio de Compensación de Seguros*, for providing the indemnity database. The cession of the oceanographic data is acknowledged to *Puertos del Estado*, a public entity dependent on the Spanish Government. We also thank three anonymous reviewers for their comments, which have helped us to substantially improve the paper.

**Conflicts of Interest:** The authors declare no conflict of interest.

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
