# Peer review of "Analysis of Ocean Parameters as Sources of Coastal Storm Damage: Regional Empirical Thresholds in Northern Spain"

_climate, doi:10.3390/cli10060088_

Round 1

Reviewer 1 Report

Please find my comments attached.

Reviewer 2 Report

This manuscript tries to identify the forcing mechanisms on storm coastal direct damage in Northern Spain. The research topic is important and interesting. However, I found this manuscript has several major limitations:

1. It is not clear the direct coastal damages and associate indemnities discussed in this manuscript are due to oceanic forcing mechanisms alone or the combination of oceanic forcing and meteorological mechanism (e.g., wind damage, pluvial and fluvial flooding). If it is the latter case, I feel only looking at the oceanic forcing mechanisms is not sufficient because it may only account for a portion of the damage.

2. Methodology: storm track and its landfall/hit location can be also important. I didn't see they were mentioned in the manuscript.

3. Most figures and tables need to be improved.

4. The overall structure of this manuscript is not well organized and difficult to follow.

5. Conclusions: I feel that the authors could just focus on the 10 major storms that accounted for the majority of the monetary loss (98.6) instead of all of them. By focusing on these major storms, the authors may do a clearer and more in-depth analysis.

5. 

Reviewer 3 Report

In the study, the authors analysed damages that were caused by coastal storms at the eastern Cantabrian coast within the years of 1996-2016. Although the study is important for coastal protection I have a couple of remarks which may improve the manuscript First of all,  prepare a good figure of the study area, tick the Iberian Peninsula, geographical coordinates. Moreover write a brief description about climate and hydrology of the examined area, present storms that occurred in 1996-2016. Parameters of wave storms might be very interesting for a potential reader. All the abbreviations should be explained in the text. Describe in detail indemnities paid by insurance companies after the most severe storms. In my view the chapter dealing with data and methods should be presented more precisely.  

Page 4, Figure 2- improve quality of the map presented in Figure 2. Discuss your results presenting a case study from the most devastating coastal storm in the examined period. Draw precise conclusions from your work.

Round 2

Reviewer 1 Report

Attached please find my last comments (minor revision needed). 

Author Response

We would like to thank you for the work you have dedicated to us. Your comments have undoubtedly improved our work.

Reviewer 2 Report

I wanted to thank the authors for carefully considering my comments and making significant improvement to the manuscript. The revised version is much improved and I'm happy with its current form.

Author Response

(The authors gave the same response as above.)

Reviewer 3 Report

Dear Editor

The ms was improved significantly and it is worth publishing

Author Response

(The authors gave the same response as above.)
